# Combining predictive models with future change scenarios can produce credible forecasts of COVID-19 futures

**Ken Newcomb**, **Shakir Bilal**, **Edwin Michael***

Global Health Infectious Disease Research, University of South Florida, Tampa, FL, United States of America

\* emichael443@usf.edu

**Data Availability Statement:** The source code and data used to produce the results and analyses presented in this article are available from GitHub repository: https://github.com/EdwinMichaelLab/COVID-FL-Vaccination.

## Abstract

The advent and distribution of vaccines against SARS-CoV-2 in late 2020 was thought to represent an effective means to control the ongoing COVID-19 pandemic. This optimistic expectation was dashed by the omicron waves that emerged over the winter of 2021/2020 even in countries that had managed to vaccinate a large fraction of their populations, raising questions about whether it is possible to use scientific knowledge along with predictive models to anticipate changes and design management measures for the pandemic. Here, we used an extended SEIR model for SARS-CoV-2 transmission sequentially calibrated to data on cases and interventions implemented in Florida until Sept. 24th 2021, and coupled to scenarios of plausible changes in key drivers of viral transmission, to evaluate the capacity of such a tool for exploring the future of the pandemic in the state. We show that while the introduction of vaccinations could have led to the permanent, albeit drawn-out, ending of the pandemic if immunity acts over the long-term, additional futures marked by complicated repeat waves of infection become possible if this immunity wanes over time. We demonstrate that the most recent omicron wave could have been predicted by this hybrid system, but only if timely information on the timing of variant emergence and its epidemiological features were made available. Simulations for the introduction of a new variant exhibiting higher transmissibility than omicron indicated that while this will result in repeat waves, forecasted peaks are unlikely to reach that observed for the omicron wave owing to levels of immunity established over time in the population. These results highlight that while limitations of models calibrated to past data for precisely forecasting the futures of epidemics must be recognized, insightful predictions of pandemic futures are still possible if uncertainties about changes in key drivers are captured appropriately through plausible scenarios.

## Introduction

The steady pace of vaccine roll outs against severe acute respiratory syndrome coronavirus 2 (SARS-CoV-2) beginning from late 2020 had raised hopes that the pandemic may soon be controlled in many economically advanced countries by as early as late 2021 or the beginning of 2022. This optimism was buttressed by falling cases and hospitalizations observed or

**Funding:** The author(s) received no external specific funding for this work.

**Competing interests:** The authors have declared that no competing interests exist.

reported during the fall of 2021 in countries and settings that had vaccinated the largest shares of their populations by that time [1, 2], and by the expectation that the available vaccines were protective against both infection and the development of clinical symptoms requiring hospitalizations from the then dominant virus variants [3–5]. It was also thought that population immunity was approaching herd immunity levels in many of these settings such that it left behind declining fractions of susceptibles that would be available to re-ignite high intensity community outbreaks [6].

The expectation that the pandemic may have reached a turning point and that we may be entering its ending phase at least in countries and settings that have been able to vaccinate a large fraction of their populations was of course dashed by the advent of the omicron variant during the winter of last year that saw the largest waves of cases observed throughout the course of the pandemic in many parts of the world [7–9]. This dramatic resurgence of the pandemic following the decline in cases observed after the delta wave has had two consequences. First, it reinforced realization of the forbidding complexity of the transmission dynamics of SARS-Cov-2, and the continual need for refining knowledge regarding the effects of changes in the biological, behavioral, and epidemiological processes that affect disease transmission at different stages of the spread of the pandemic in a given setting [10–14]. Such gaps in knowledge after the delta wave included gaining a better understanding of the likely outcomes arising from continuance of transmission among unvaccinated subpopulations for pandemic persistence as well as for the emergence and spread of virus mutants [15, 16], uncertainties regarding the protective efficacy of individual vaccines against different variants, including the impacts of breakthrough infections among vaccinated individuals [17], and critically the impact of a less than permanent operation of anti-viral immunity [18–20]. Further, understanding how these factors interact among themselves and with reductions in public observances of social mitigation measures [21] were also considered essential for determining if we will be able to halt virus transmission using mass vaccinations, or if we should expect to see additional resurgences of the pandemic going forward even in populations that had received high levels of vaccinations [22].

The resurgence of the pandemic over the winter of 2021 and early 2022 has, secondly, also refocused attention on whether it is possible to forecast the future stages of a contagion reliably using mathematical models [10, 11]. Thus, while some workers have highlighted the difficulty of anticipating and accommodating novel, unknown, or previously unsuspected changes in the drivers of future viral transmission to allow the making of reliable long-term projections by models [10, 11], others have pointed to the value of these models as tools for being able to integrate information on the diverse structures and processes related to transmission dynamics in order to propagate forecasts that are more accurate than predictions afforded by common sense alone [12, 23]. Such assessments have also, for example, pinpointed the need for continual model refinement and for the use of data for making predictions to counter the effects of changes in local risk factors [12, 14, 24]. These studies ultimately suggest that, as for other studies investigating socio-ecological futures, the better use of models for forecasting the plausible futures that could be followed by the pandemic is to combine simulations within a scenario framework in order to focus on explorations of possible trajectories in the evolution of the system as a result of changes predicted for key drivers rather than employing them to make precise predictions about the extent or duration of disease burdens [25, 26].

Here, our goals are twofold. First, we consider the ability of our previously developed data-driven socio-epidemiological SEIR-based COVID-19 model [21, 27, 28] that includes vaccination, social mitigation and variant-specific transmission dynamics and fit to sequential data, in order to determine the ability of this type of compartmental model to predict the future paths that could be followed by the COVID-19 pandemic. We do this by using the model fitted to

data prior to the omicron-related pandemic resurgence observed in the state of Florida and determining its predictability for the actually realized omicron wave that followed in the state under different variant emergence, transmissibility, social mitigation and vaccination scenarios. This allowed us to inspect the capacity of such models, whose parameters are essentially conditioned on past data, to capture and forecast the outcomes of novel, generally unknowable, system risks. At its philosophical core, therefore, our goal was to assess where this type of model that is inductively identified using past and present data may have the exchangeable information required within its structure and parameters to allow insights into possible, even anticipated, futures that might be followed by the pandemic in a particular location [25, 26, 29, 30]. We show in this regard that while additional information regarding the timing of variant emergence and transmissibility was critical for propagating the realized resurgence of the pandemic due to omicron, our data and scenario-based model can nonetheless offer valuable qualitative insights into how the pandemic might behave post-omicron under conditions of impermanent immunity and the emergence and spread of new escape mutants. More generally, the results of this work imply that if procedures to prevent significant depletions in parameter variance are implemented during model updating [31–33], then models like ours that are fitted sequentially to data and used in combination within a scenario-based framework can offer a useful tool for generating projections that can allow reasonable assessments to be made regarding the plausible future paths that may be followed by the current pandemic.

## Methods

### Basic SEIR model

Our previous data-driven SEIR-based COVID-19 model was extended to include the dynamics of imperfect vaccines, new genetic variants, and impacts of social mitigation measures to perform the present simulations [21, 27, 28]. Briefly, the basic model simulated the course of the pandemic in a particular setting via the adaptive rate of movement of individuals through various discrete compartments, including different infection and symptomatic categories as well as immune, vaccination and death classes. The basic model includes three COVID-19 variants (alpha, delta, and all others), and is further extended to simulate the emergence of new variants. We also assume that the modelled population is closed and the population size remains constant over the duration of the simulations reported here. The full set of equations and description of the model are provided in a public GitHub repository (see link given below). Here, we outline how the extensions are implemented in the basic model framework.

### Vaccination/Breakthrough dynamics

The impact of vaccinations is simulated by first moving susceptible Individuals into the vaccinated compartment (V) according to the reported daily vaccination rate ($\xi_v$.). We also note that as recipients of vaccines are not first tested to determine their immune status, vaccines will be provided also to individuals who are immune (R). We capture this wastage via the term $\xi_v(1-R)$; this implies that such vaccine wastage will scale with the numbers of individuals who are in the R classes at any given time. Vaccinated individuals are then moved from the V class to the booster class (2nd dose, B) at a daily rate approximating a 6-week interval between vaccine doses. A further third dose of vaccine is given 1 month after waning of immunity, as discussed below, to generate a third vaccine class, T. The vaccines are also assumed to be imperfect, which thus allow for breakthrough infections in some vaccinated individuals [34]. To model this, vaccinated individuals are simulated to acquire infection following effective contact with individuals in the various I classes, at reduced rates controlled essentially multiplying the transmission rate (β) by the vaccine efficacy factor (1- ε) that is assumed to vary

**Table 1. Vaccine efficacies used in the simulations.**

|  | 1st dose efficacy | 2nd dose efficacy | Waned Immunity | 3rd dose efficacy |
|---|---|---|---|---|
| **Original Variant** | 75% | 90% | 80% | 99% |
| **Alpha Variant (B.1.1.7)** | 70% | 85% | 75% | 99% |
| **Delta Variant (B.1.617.2)** | 65% | 80% | 70% | 99% |
| **Omicron Variant (BA.1)** | 35% | 45% | 35% | 70% |
| **Omicron Subvariant (BA.2)** | 25% | 35% | 25% | 60% |

between vaccine doses and variants (see Table 1). Average vaccination rates estimated from the last 7 days of the vaccination data up to Sept. 24th 2021 in Florida were used to simulate into the future. The vaccine efficacies used in this study [35–37] are given in Table 1 below. It has been demonstrated [36] that the efficacy of a two-dose regimen of mRNA vaccines can reach levels after 2 months, but decays to 67–80% after 7 months. In addition, protection against the Delta and Omicron variants is also shown to be significantly lower than to the original and alpha variants [36, 37]. We thus derived and used efficacy values mirroring these patterns in this study.

Waning of vaccine-induced immunity was explored by allowing individuals in the B (or 2nd dose) state to move into a reduced efficacy state (W (see Fig 1 and Table 1)) over 5 months. The waning of natural immunity was also explored over 1 year, 2.5 years, and 5 years. In this case, individuals are simply moved from the recovered state back to into two fully susceptible states (Fig 1). The first of these states (S) will be replenished with individuals who recover from naturally acquired infection but are yet to vaccinated. We, however, consider that these individuals will be willing to be vaccinated following their recovery. The individuals who experience breakthrough infections after being vaccinated and subsequently become infected and recover are moved into vaccinated-susceptible class (VS). Since the time between the 1st dose of vaccine and the 3rd dose (8 months) is significantly shorter than the length of natural

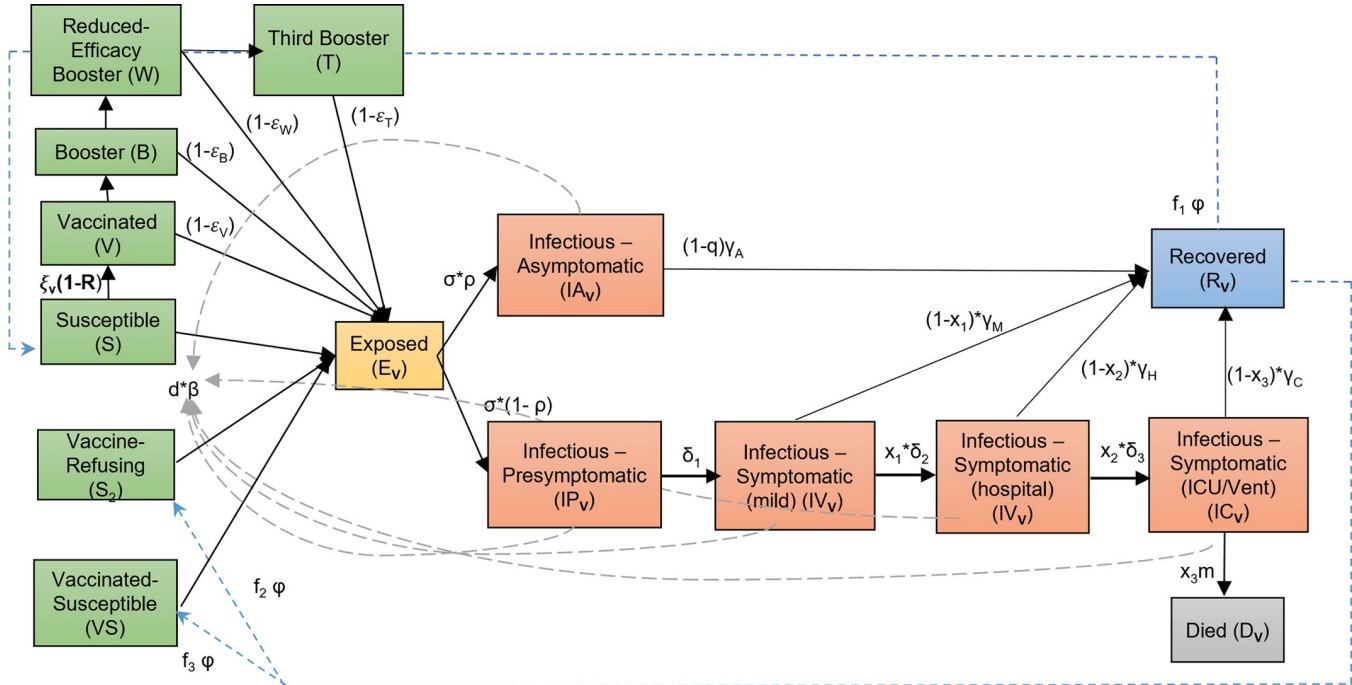

**Fig 1. Expanded SEIR model flowchart.** Each compartment represents a distinct state of infection, with arrows indicating flows between compartments.

immunity (1 to 5 years, depending on the scenario), these individuals are assumed to be fully vaccinated after recovering from natural infection. Finally, we model a small proportion of the population as vaccine-refusing, represented by the S2 class. (9% estimated for FL (https://vaccine-hesitancy.healthdata.org/).

## Adding variants

Since the alpha and delta variants were dominant at various points during the pandemic until Fall 2021, we decided to model them explicitly. This was performed using the framework described by Davies and coworkers [38], which essentially uses individual strain models fitted to longitudinal viral sequence-derived variant case data to estimate variant-specific transmission rates. We also assume that previously dominant strains provide some level of cross-protection against emerging new variants [38, 39]. Each variant (alpha, delta, and all others) therefore has a corresponding exposed (E), infectious (IA, IP, IM, IH, IC), recovered (R), and death (D) compartments along with estimated variant-specific transmission rates, and a cross-immunity parameter by which immunity generated by individuals against previous strains confers a 10% level of protection against new emerging variants (see GitHub Repository for equations). The proportionate data on each variant was taken from the Helix COVID-19 Surveillance Dashboard (https://www.helix.com/pages/helix-covid-19-surveillance-dashboard), which compiles longitudinal genetic surveillance data from each state. These proportions along with Florida's reported case data were used to estimate the corresponding time-dependent changes in variant-specific cases, which allowed for the model to be fit to the cases of each variant over time. In addition, the impact of future emergences of new SARS-CoV-2 variants with higher transmissibility and immune evasion was also explored. To do this, we first computed the average transmission rate of the original, alpha, and delta variants, and introduced a new 4th variant with 50% and 100% increases in the transmission rate. We considered the possibility of emergence of these putative 4th variants on Nov. 1st 2021 and Dec. 1st 2021 to investigate if our model calibrated to data up to Sept. 24th 2021 was able to reproduce the omicron wave observed. A similar simulation strategy was also employed to evaluate the impact that introductions of a 5th future variant in 2022 would have on the path of the pandemic.

## Sequential model calibration and selection

Calibration of the model to capture the transmission conditions of Florida was performed by fitting the SEIR model sequentially to daily confirmed case and mortality data assembled from the start of the epidemic until September 24th, 2021, as provided by the Coronavirus App (https://coronavirus.app). Similar sequential fitting procedures have been used in other simulation studies of COVID-19, in order to capture the rapidly changing transmission conditions, while at the same time retaining some information regarding the past [40]. A 7-day moving average is applied to the daily confirmed case and death data to smooth out fluctuations due to COVID-19 reporting inconsistencies. A sequential Monte Carlo-based approach was used for carrying out the updating of the model by sampling 20,000 initial parameter vectors initially from prior distributions assigned to the values of each parameter for every 10-day block of data [24]. An ensemble of 250 best-fitting parameter vectors, based on a modified Normalized Root Mean Square Error (NRMSE) between predicted and observed case and death data, is then selected for describing these 10-day segments of data as described previously [24]. Updating of parameters is accomplished by using the best-fitting ensemble of parameter posteriors as priors for the next 10-day block, and the fitting process is repeated. Importantly, we also inserted 25% of parameter vectors drawn randomly from the initial prior distributions into the priors at each updating episode to avoid parameter variance depletion [31–33, 41].

## Estimating social mitigation levels

The strength of social distancing measures imposed by authorities to limit contacts is captured through the estimation of a scaling factor, $d$, which is in turn multiplied by the transmission rate, $\beta$, to obtain the population-level transmission intensity operational at any given time in a population (Fig 1). This factor accounts for the effects of mask wearing, reductions in mobility and mixing, working from home, and any other deviations from the normal social behavior of a population prior to the epidemic. To set the priors for the social distancing parameter $d$, Google Trends search data was leveraged. The Google Trends API provides a normalized measure of web searches for the phrases "covid" and "covid mask" in Florida on a particular day, which we expect to correlate with levels of social distancing followed by the population. We used a range of values 10% above and below the average of the Google Trends values for the above phrases to serve as the priors for the social distancing parameter, $d$, before each model updating period.

## Full model

The coupled differential equations governing the evolution of the full extended system, the model code used to perform the simulations, and all prior and posterior fitted parameter values for the best-fit models calibrated to data to Sept. 24[th] 2021 are given in the Table provided at https://github.com/EdwinMichaelLab/COVID-FL-Vaccination. The ensemble of best-fitting models obtained from the sequential model calibrations was used to forecast the impacts of the various future scenarios related to immunity durations and the advent and spread of new variants explored in this paper. Fig 1 provides a flowchart of the full structure of the extended SEIR model described above.

## Fade-out probability calculations

The probability of pandemic fade-out was assessed via simulation as follows. First, we used the ensemble of models that best fit the latest data (Sept. 24[th] 2021 in this case) to generate forward trajectories for the pandemic. For a given timestep, we then computed the fraction of those trajectories that showed strictly decreasing cases into the future. A trajectory is considered decreasing if their predicted cases are currently higher than they will be one week in the future; this weekly assessment also ensures that daily fluctuations in cases are ignored. The fraction of such trajectories is used directly to calculate the probability of elimination of the pandemic over the chosen timestep. This analysis was performed for the case of continued social distancing measures and vaccination, and under the conditions of full release of social measures. These estimations of fade-out probabilities were carried out for the case of long-term immunity and for immunity that persisted for at least 2.5 years (see results).

## Estimation of population immunity under varying durations of immunity

The level of population immunity attained, and the date at which maximal immunity, is achieved were also estimated through simulation. The fraction of total recovereds (R) predicted by the model at a given time includes both the fraction vaccinated and the fraction recovering from natural infection (Fig 1). This allowed us to estimate the change in levels of immunity due to natural infection versus that arising from vaccination. The impact of waning of immunity is similarly evaluated by forecasting the changes in the total immune fractions over time under conditions of continuance with extant social measures and a full release of these measures.

## Results

### Estimation of variant-specific models and consequences of long-term immunity

Fig 2A shows the course of the COVID-19 pandemic in terms of daily confirmed cases (solid lines) in Florida from early March 2020 when it first emerged to cases reported on Sept. 24[th] 2021. These data depict the typical wave-like trend expected for daily cases of pandemics due to implementation of temporally varying public interventions and social behavioral changes that are not strong or applied long enough to bring about the permanent breakage of community transmission [42, 43]. It also shows the advent and spread of SARS-CoV-2 variants with the alpha variant first emerging in late December 2020 and the delta variant first appearing in June 2021 before becoming the pre-dominant variant from mid-July 2021 to Fall 2022 in the state. The dotted lines show the ability of our variant-specific data-driven model to faithfully capture the observed dynamics of these viral variants, including the overtaking of the alpha variant by the more transmissible delta variant (estimated rate of transmission being 1.8 for the delta variant compared to 1.1 and 1.2 for the alpha and original variants respectively (Fig 2A)).

We examined the effect of long-term immunity on the future path (to end of 2022) that may be followed by the pandemic by combining the models parameterized using data to Sept. 24[th] 2021 with scenarios that included the operation of long duration immunity but which differed in the levels of social protective measures and vaccinations implemented. These simulations firstly show that the delta variant-induced wave of the pandemic peaked on Aug. 26[th] 2021 at 22,400 median daily cases in line with case reports (Table 2), with cases declining thereafter under the levels of social mitigation (23%) and vaccinations (20,000/day) observed to Sept. 24[th] 2021 (Fig 2B). If immunity to SARS-CoV-2 is long-term, the predictions for this scenario also indicate that the pandemic will fade out in early 2022 (see below). The solid blue curve shows that fully releasing social protection measures from Sept. 24[th] 2021 will result in only a small increase in cases (over those produced under maintaining the social protective measures observed around Sept. 24[th] 2021), which will subsequently decline to small levels from July 2022. It is notable that in direct contrast, if social mitigation measures had been released on Mar. 1[st] 2021, a major spike in cases would have occurred (blue dashed curve). Increasing the vaccination rate 1.5x from Sept. 24[th] 2021 (to approximately mimic the school vaccinations that were being proposed then) under maintenance of the then observed social protective measures would have resulted in lower future cases but not significantly so compared to the predictions for the pandemic future given continuance with these social measure/ vaccination levels (green dashed curve). Releasing such social mitigation measures fully while increasing the vaccination rate by 1.5x, however, would have resulted in an increase in cases but this increase would only be slightly lower than that predicted for when the observed vaccination rate is continued (dashed magenta curve and Table 2).These results indicate that releasing social measures fully and increasing the vaccination rate from Sept. 24[th] 2021 would have only a moderate impact on the future course of the pandemic under conditions of permanent immunity.

These results highlight the impact of the changed immunity or conversely the susceptibility landscape that was established in Florida by Sept. 24[th] 2021, whereby up to 90% of the population was predicted to have developed immunity to the virus from both infection and vaccination (Fig 3A), with herd immunity (estimated at 91% under the operation of long-term immunity) forecasted to be reached as early as Nov. 22[nd] 2021. The fraction of the population immune was much lower in March 2021 (approximately just 30–35%), and thus formed the primary reason for the large spike in cases predicted for a full release of social measures at that stage compared to the small increase in cases resulting from the release of these measures from

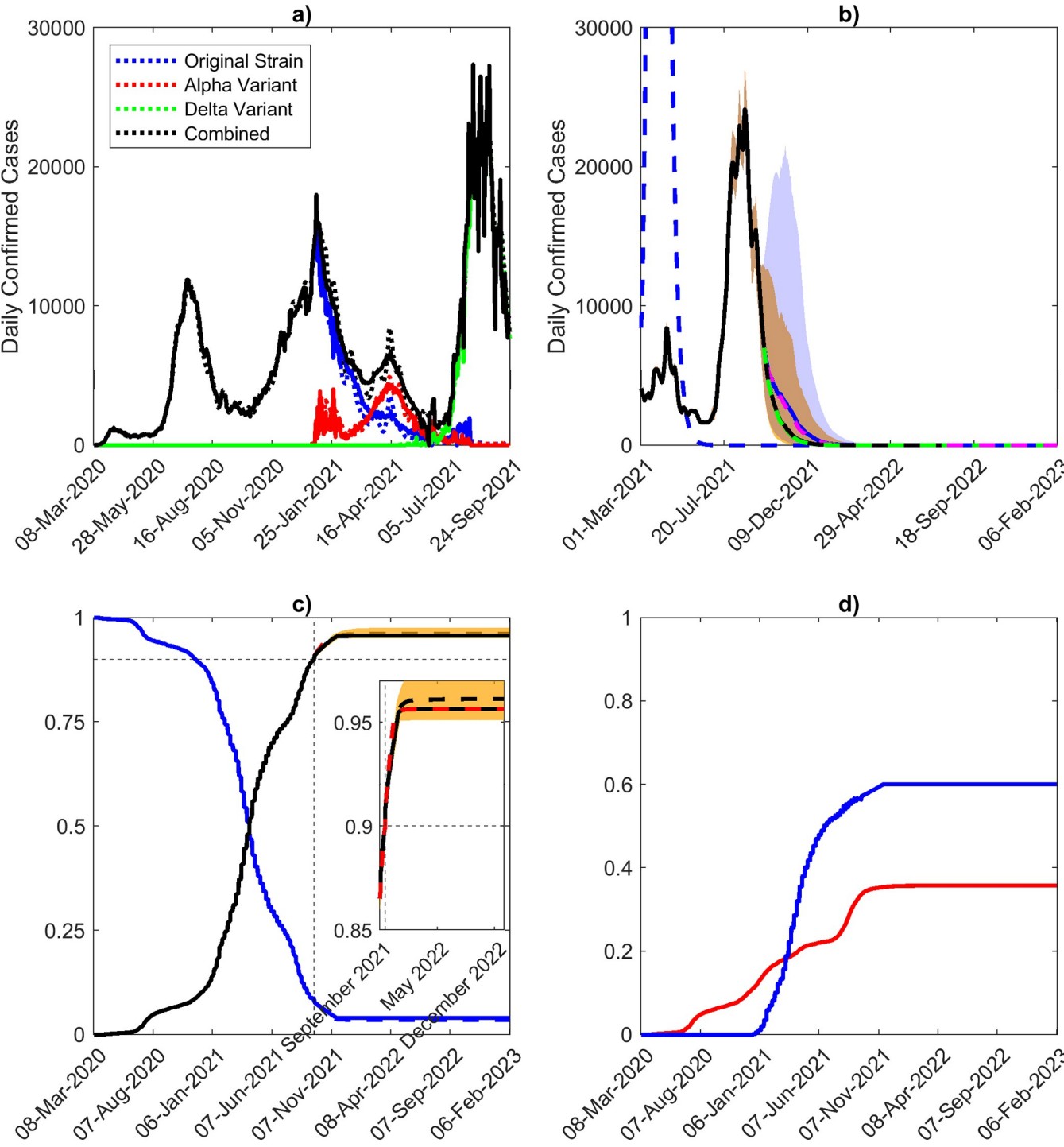

**Fig 2. Model fits and long-term predictions.** a) Median model fits to the 7-day moving average of daily confirmed case data, along with short-term predictions, assuming long-term immunity. The cases due to the alpha, delta, and other variants are given in red, green, and blue, respectively. b) Long-term forecasts of daily confirmed cases until February 2023. The 7-day moving average of confirmed case data is given in red. The median model predictions given estimates of social distancing measures and vaccination rate as of Sept. 24[th] 2021 are given by the black curve, while the model predictions given a full release of social distancing measures are given by the blue curve. The yellow and blue shading represent the 90% confidence intervals for these two scenarios. If the vaccination rate is increased by 1.5x, the median model prediction given social distancing measures estimated on Sept. 24[th] 2021 is given by the green dashed curve, while in the case of full release of social measures, it is given in magenta. The blue dashed line represents a full release of social measures on Mar. 1[st] 2021, when the fraction of immunity was much lower than present, which produces a peak of 250,000 daily confirmed cases (assuming social measures are not modified). c) Total proportion susceptible (blue), and total immune (black) over time. The proportion immune given estimates of social measures and

vaccination as of Sept. 24[th] 2021 is given by the solid black line, while a full release of social measures is shown as a dashed black line. If the vaccination rate is increased by 1.5x, the proportion immune is represented by the red dashed line. The 90% confidence interval is shown as a yellow band. As of Sept. 24[th] 2021, 9% of the population were susceptible, while 90% were immune. d) Proportion of the population with natural immunity (red) along with the proportion of the population with vaccine-conferred immunity (blue). As of Sept. 24[th] 2021, the fraction of the population with natural immunity is 34%, while the fraction with vaccine-induced immunity is 56%.

Sept. 24[th] 2021. Interestingly, Fig 2D shows that by that stage, vaccinations had contributed to 56%, while naturally acquired immunity (ie immunity through infection) comprised 34% of the population immunity generated against SARS-CoV-2 in Florida.

The forecasts of the models updated using data to Sept. 24[th] 2021 for the mix of social measures/vaccination levels investigated for hospitalizations and deaths are also shown in Fig 3 and Table 2. The results corroborate the findings for daily confirmed cases in that while no peaks would have been seen or would emerge for the scenarios that maintained social measures into the future irrespective of vaccination rate, whether followed at the then rate (~20,000 doses per day) or 1.5x the observed rate, new small peaks could develop in the future (in Nov. 2021) for the two scenarios in which social measures are fully released from Sept. 24[th] 2021 (Table 2; Fig 3). The impact of increasing vaccinations by 1.5x of the then achieved rate would have only a small effect on these predicted peaks. Note although increases in these clinical outcomes are predicted in the future for these scenarios, the daily numbers at peak will be substantially lower than those which occurred (and predicted) in Aug. 2021 for the base scenario investigated, viz in which social measures and vaccinations are held at their Sept. 24[th] 2021 levels.

**Table 2. Peak cases, hospitalizations, deaths.**

| With Sustained Long-Term Immunity | | | | |
|---|---|---|---|---|
| Scenario | | Cases | Hospitalizations | Deaths |
| Estimated Social Measures as of Sept. 24[th], 2021 | Estimated Vaccination Rate as of Sept. 24[th], 2021 | No peak | No peak | No peak |
| Estimated Social Measures as of Sept. 24[th], 2021 | 1.5x Vaccination Rate | No peak | No peak | No peak |
| Full release of social measures | Estimated Vaccination Rate as of Sept. 24[th], 2021 | No peak | No peak | No peak |
| Full release of social measures | 1.5x Vaccination Rate | No peak | No peak | No peak |
| With Waning of Immunity over 1yr | | | | |
| Scenario | | Cases | Hospitalizations | Deaths |
| Estimated Social Measures as of Sept. 24[th], 2021 | | 16,900 cases on November 20[th], 2021 | 11,000 beds on November 23[rd], 2021 | 200 deaths on November 28[th], 2021 |
| Full Release of Social Measures | | 54,000 cases on November 1[st], 2021 | 32,000 beds on November 11[th], 2021 | 560 deaths on November 13[th], 2021 |
| With Waning of Immunity over 2.5 yrs | | | | |
| Scenario | | Cases | Hospitalizations | Deaths |
| Estimated Social Measures as of Sept. 24[th], 2021 | | 8000 cases on October 29[th], 2021 | No peak | No peak |
| Full release of social measures | | 23,000 cases on November 2[nd], 2021 | 14,000 beds on November 12[th], 2021 | 325 deaths on November 18[th], 2021 |
| With Waning of Immunity over 5 yrs | | | | |
| Scenario | | Cases | Hospitalizations | Deaths |
| Estimated Social Measures as of Sept. 24[th], 2021 | | 5000 cases on November 2[nd], 2021 | No peak | No peak |
| Full release of social measures | | 17,000 cases on November 17[th], 2021 | 12,000 beds on November 21[st], 2021 | 230 deaths on November 24[th], 2021 |

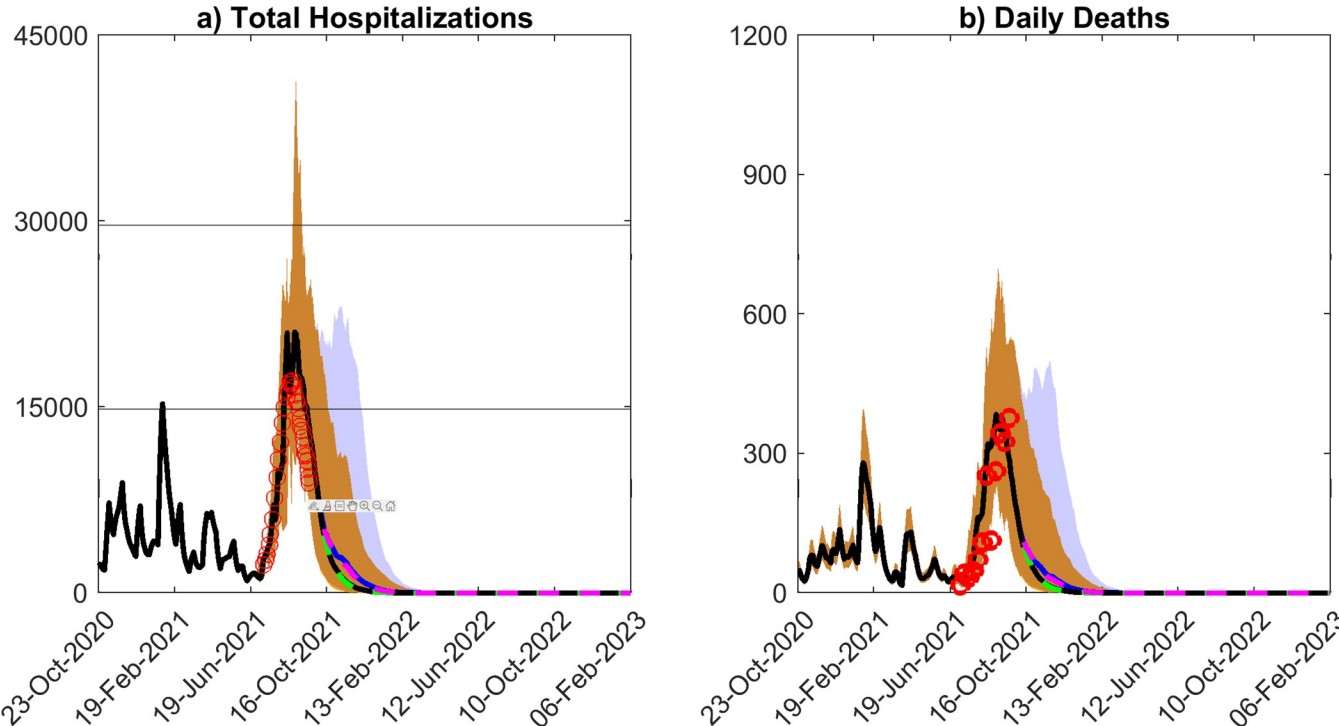

**Fig 3. Model predictions of hospitalizations and deaths.** Median predictions of a) total hospitalizations and b) daily deaths over time. Several scenarios are shown. Median model predictions given estimates of social distancing measures and daily vaccination rate as of Sept. 24th 2021 are shown by the solid black curve, while the median model predictions given a full release of social measures is shown in blue. The 90% confidence interval for these two scenarios (social measures estimated on Sept. 24th 2021, and full release of social measures) are given by the yellow and blue bands, respectively. If vaccination rate is increased by a factor of 1.5x, the median model predictions are shown by the green curve when applying estimates of social measures as of Sept. 24th 2021 and shown in magenta under a full release of social measures. The daily hospitalization and death data is shown by the red circles.

### Pandemic future under waning immunity scenarios

Fig 4 illustrates the likely paths of the pandemic if population immunity is not permanent or long-term and were to wane over durations of 1 year (fast waning), 2.5 years (moderate waning) to 5 years (semi-permanent). Forecasts are show for the situation in which estimates of social protective measures and vaccination rates as of Sept. 24th 2021 are continued in the future and when social measures are fully released from Sept. 24th 2021 onwards. We also show results for increasing the vaccination rate by 1.5x the then rate, while maintaining estimates of social measures into the future. It is immediately apparent as expected that if immunity were to wane, the pandemic will settle into a cyclical pattern of rise and depletion in cases with amplitudes (and peak cases) and inter-wave periods dictated by the duration over which immunity wanes. Sizes of the oscillating waves would decline while lengths of inter-wave periods will increase with increasing duration of immunity (Fig 4). Under a scenario where the then social measures/vaccination rate was to be maintained into the future, the pandemic will recede and remain suppressed for a long period of time and any resurgence (beyond the period of simulation shown) will be easily containable.

Full release of social measures, by contrast, can still be dangerous and could result in large resurges particularly if duration of immunity is short (eg. 1 or 2.5 years). The predicted peak cases, hospitalizations and deaths given in Table 2 for these two scenarios (ie. continuing the then vaccination rate with estimates of social measures as of Sept. 24th 2021 versus full release of social measures) further buttress this conclusion. Increasing the vaccination rate, compared

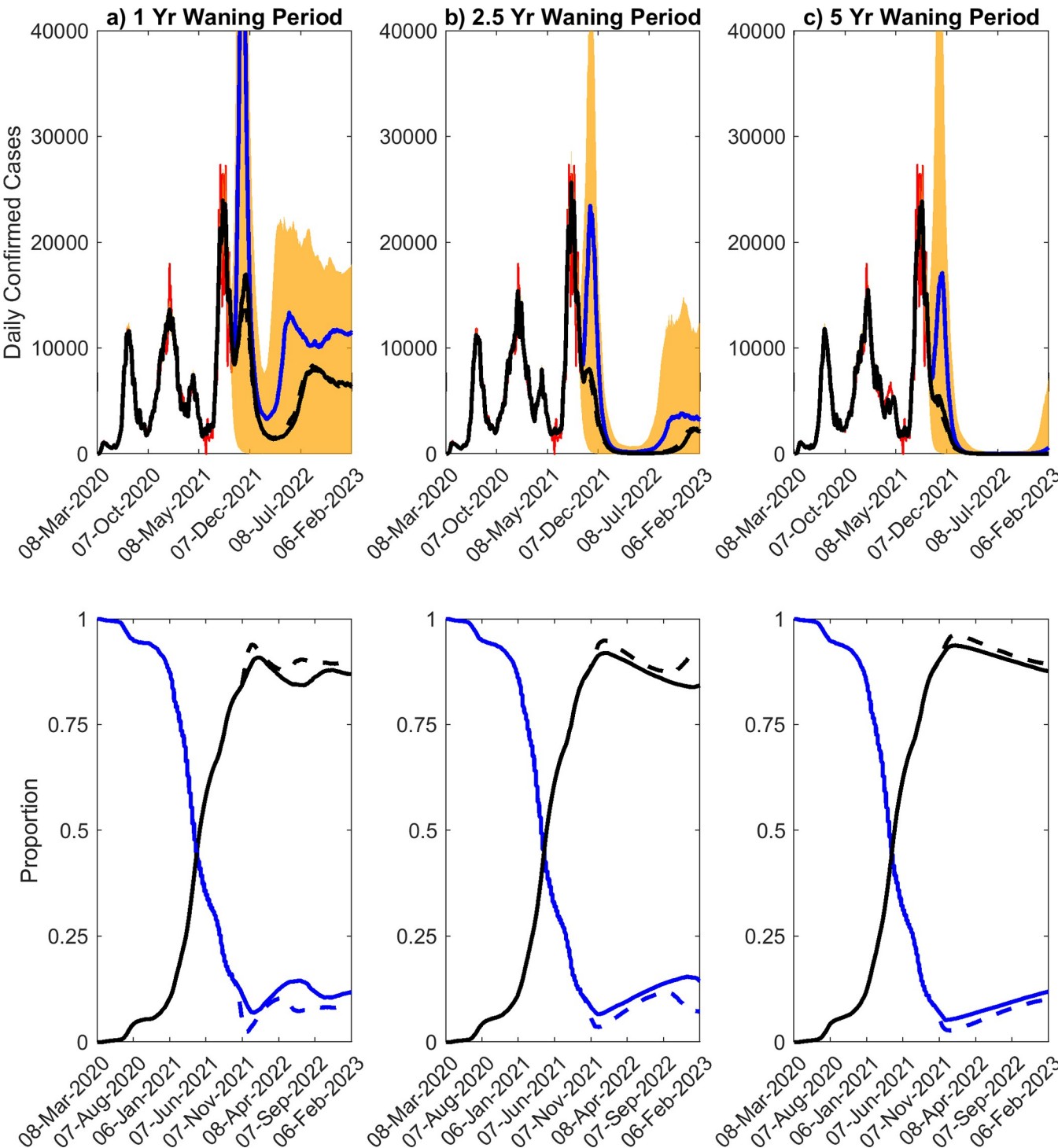

**Fig 4. Scenarios of waning immunity.** (Top row) Scenarios of waning immunity, given waning intervals of 1 year, 2.5 years, and 5 years. The median model predictions of confirmed cases given estimated social distancing measures as of Sept. 24th 2021 and release of social measures are given by the black and blue curves, respectively. The 90% confidence interval for the median case is given by the orange shading, while the confirmed case data is given in red. (Bottom row) Forecasts for total proportion susceptible (blue), and total immune (black) are shown over time. Solid curves represent the impact of maintaining social measures and vaccination rates as of Sept. 24th 2021, while the dashed curve denotes the effects for a full release of social measures.

to maintaining the rate observed on Sept. 24[th] 2021, will initially cause a decrease in daily confirmed cases, but will lead to a small peak above the cases forecasted for continuing with September 24[th] rate. This occurs only if immunity wanes quickly (over 1 year, Fig 4). However, the cumulative cases generated are reduced by increasing the vaccination rate (see Supporting information, S4 Fig), indicating that the increase in daily cases is a transient outcome of infection breakthroughs among the vaccinated individuals particularly when waning of immunity is rapid. Fig 4 clarifies the primary reasons for the oscillatory dynamics forecasted for the pandemic under conditions of waning immunity; the results show that with waning of immunity, herd immunity may never be reached leading to revivals of the susceptible fraction in the population with the negative impact on achieving population immunity and the increase of susceptibles more apparent as the duration of immunity declines.

We calculated and used the RMSE values of fits of our models to the case data observed over a 4-week period around the peak of the 4[th] (delta) wave to detect signals for the emergence and operation of waning immunity. In this approach, we assume that better fits by models with waning immunity over the model with permanent immunity may allow us to distinguish which of these types of immunity may be becoming operational and thereby offer a clue as to the likely future path that might be followed by the pandemic in Florida. Table 3 displays the RMSE values and relative errors of the fits of the models without and with waning of immunity. These show that models with waning immunity provided better fits (smaller RMSE values) and reduced the model errors more relative to the model with no waning of immunity. However, the model that gave the best fit and reduced modelling errors most was that which incorporated the moderate waning duration (2.5 years) investigated in this study, indicating that if waning of immunity is playing a role in describing the current state of the pandemic then the future path of the pandemic will follow one in which immunity may act over a relatively moderately long duration (eg. the path of the pandemic arising from immunity that wanes over 2.5 years (Fig 5)).

## Pandemic fade-out probabilities for different durations of immunity

We used projections from individual models belonging to our best fitting multi-model ensemble to calculate the probability of pandemic fade-out (see Methods). This was carried out under both the operation of long-term versus a 2.5 year waning of immunity. Fig 5A shows that if the social measures and vaccination rates operating in Sept. 2021 are maintained into the future, then we would have reached a very high probability of fade out (>99%) of the pandemic by Nov. 13[th] 2021 if immunity operated long-term. However, releasing all social measures from Sept. 24[th] 2021 under this immunity would have delayed the time to fade out of the pandemic (Fig 5A). The results show that under this scenario, fade out at the corresponding 99% probability level will now only occur just after Nov. 22[nd] 2021. If the vaccination rate, on the other hand, is raised from the Sept. 2021 level of 20,000 doses per day to 30,000 doses per day (a 1.5x increase) while the social protection observed until then is continued, the fadeout of the pandemic under such immunity would be achieved on Nov. 8[th] 2021, and Nov. 19[th]

**Table 3. RMSE and relative error of models with and without waning of immunity, from July 27[th], 2021 to Sept. 16[th], 2021 (delta wave peak).**

| Model | RMSE | Relative Error |
|---|---|---|
| No Waning | 40.0 | 0 |
| 1yr Waning | 41.7 | +4.2% |
| 2.5yr Waning | **36.0** | -10.0% |
| 5yr Waning | 39.8 | -0.5% |

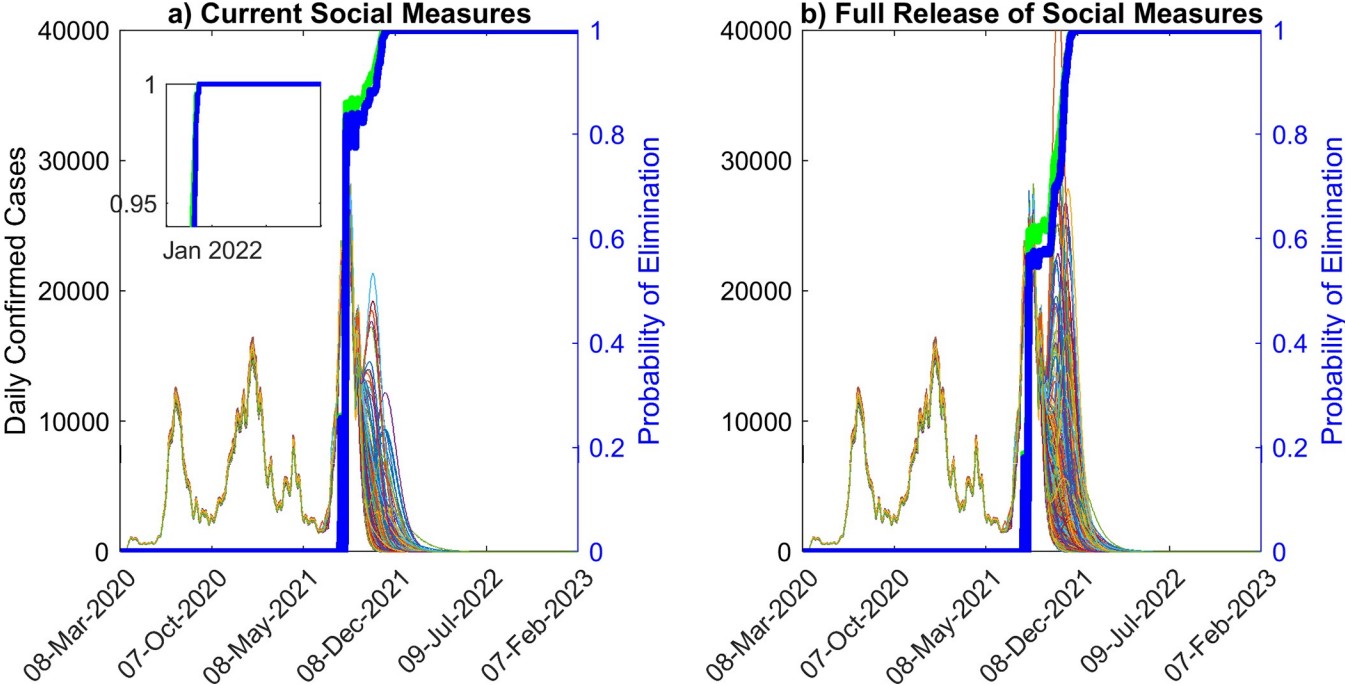

**Fig 5. Model ensemble and probability of elimination over time, given long-term immunity.** Ensemble of predictions of daily confirmed cases, and probability of elimination over time, assuming long-term immunity, given a) estimated social measures as of Sept. 24[th], 2021 and b) after the full release of social measures. The predictions of the model ensemble (250 in total, see Methods) are represented by the thin curves in the background of the figure. Given the estimated social measures and vaccination rate as of Sept. 24[th], 2021, the probability of fadeout is given by the blue curves, whereas increasing vaccination to 1.5x is presented by the green curves, respectively. If estimated social measures are continued along with the vaccination rate as of Sept. 24[th] 2021, 99% probability of elimination will be achieved on Nov. 13[th] 2021, while if social measures are fully released, 99% probability of elimination will be achieved on Nov. 22[nd] 2021. With a 1.5x increase in vaccination, the corresponding 99% probability of pandemic fadeout will be achieved on Nov. 8[th] 2021 and Nov. 19[th] 2021 for continuing with estimated social measures and given a release of social measures, respectively. Even though there is a significant probability of resurgence given a full release of social measures, the size of the wave is likely to be very small.

2021 in the case of full release. These results again indicate that increasing vaccination by 1.5x the rate observed in Sept. 2021 would have resulted in only a slightly earlier fade-out of the pandemic under conditions of permanent immunity. This is primarily because very high levels (90%) of population immunity had already been established in the state's population by Sept. 24[th] 2021 (Fig 2).

Our investigation of fadeout probabilities if immunity were to wane (here modelled for 2.5 years) indicate a dramatically different pattern to that predicted above for the condition of permanent immunity. The results are depicted in Fig 6, and show firstly that if immunity to SARS-CoV-2 is not permanent then compared to the scenario of permanent immunity (Fig 5), fadeout probabilities for the pandemic will not only remain significantly low (eg. reaching 50% only in 2023), but will also not reach the level smoothly as observed for the case of permanent immunity. As shown in Figs 4 and 6 this outcome is instrumental to causing resurgences of the pandemic until eventually a steady endemic state is achieved over the longer-term.

It is also instructive to compare the fadeout probabilities calculated at the time when the model was last updated (ie Sept. 24[th] 2021) for the long-term versus impermanent immunity scenarios to serve a metric for assessing the likely future paths of the pandemic. Our calculations showed that if long-term immunity was operation, then the fadeout probability achieved given the social measures practiced up to Sept. 24[th] 2021 was close to 82.4%, whereas it was only 20.4% at that time under the same social conditions when immunity lasted for up to 2.5 years. This indicates that Florida was still at high risk for future pandemic resurgences from

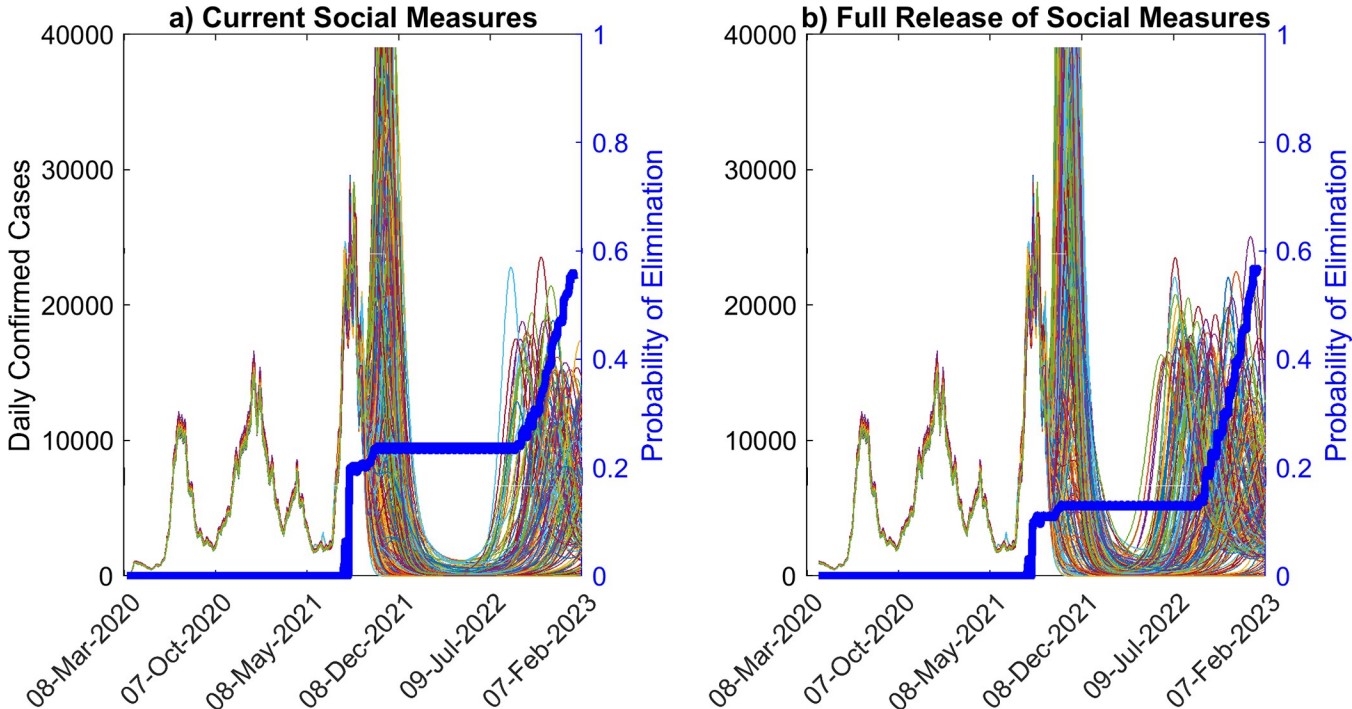

**Fig 6. Model ensemble and probability of elimination over time, given waning of immunity.** Ensemble of predictions of daily confirmed cases, and probability of elimination over time, assuming waning of natural immunity over 2.5 years, given a) estimates of social measures as of Sept. 24th 2021 and b) after the full release of social measures. The predictions of the model ensemble (250 in total, see Methods) are represented by the thin curves in the background of the figure. For the estimated social measures and vaccination rate, the probability of fadeout is given by the blue curves. For these scenarios, the probability of elimination does not reach 50% until 2023.

Sept. 24th 2021 if immunity to the virus was impermanent, and further that its population would also be significantly vulnerable to the emergence of more transmissible new variants.

## Impact of new variants

Our data-driven model is based on sequential assimilation of information from longitudinal case/death data to both minimize initial condition uncertainty [30, 44] and to incorporate temporal changes in external drivers of transmission (containment by social measures, vaccinations, and arrival of variants). Such constraining of model parameters can preserve internal stability in the predictions [30], but can lead to parameter depletion or invariance when models are sequentially updated through time whereby posteriors from a previous fit is used as priors for subsequent fits [32, 33, 41]. This can reduce the capacity of sequentially fitted models to capture the effects of novel, anticipated, drivers, such as impacts of the advent of future variants. We have included a parameter variance maintenance mechanism by means of blending in 25% of values from the original priors set for each parameter into the sequential priors used to update our model to counter the problem of parameter depletion, and here, we inspected the utility of this approach for the ability of our data-model framework for forecasting the impact of the arrival of a new mutant post Sept. 24th 2021 in Florida. We used the data depicting the omicron wave that emerged mid Dec. 2021 and rose rapidly to peak around Jan. 9th 2022 in Florida to determine if and under what conditions our model that was estimated using data to Sept. 24th was able to forecast this future wave in this investigation. Four emerging variant scenarios were simulated, viz. a new variant that is either emergent on Nov. 1st 2021 or

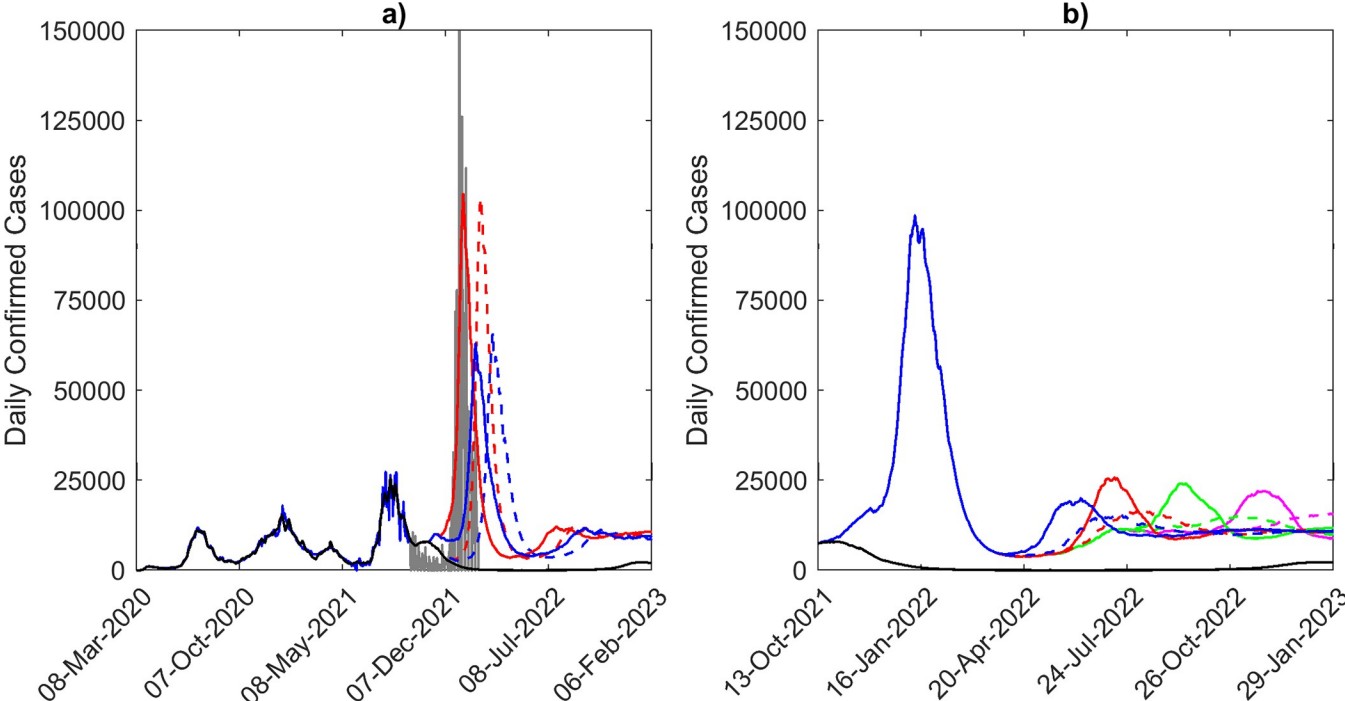

**Fig 7. Simulations of emerging variants.** a) Median model predictions (black) fit to the 7-day moving average of daily confirmed cases in Florida to Sept. 24[th] 2021 (blue), given waning of immunity over 2.5 years. Confirmed case data from Sept. 24[th] to Feb. 14[th] 2022 is shown by the gray curve. Four scenarios of the emergence of a 4[th] variant are shown: 2x transmissibility of the original variants, emerging on November 1[st] (red, solid line), 1.5x transmissibility, emerging on November 1[st] (blue, solid line), 2x transmissibility, emerging on December 1[st] (red, dashed line), and 1.5x transmissibility, emerging on December 1[st] (blue, dashed line). b) Simulations of a 5[th] variant, emerging on Mar. 1[st] 2022 (blue), May 1[st] 2022 (red), July 1[st] 2022 (green), and Sept. 1[st] 2022 (magenta). Solid lines represent the case of a 5x increase in transmissibility compared to the original variant, while the dashed lines represent a 3.5x increase in transmissibility.

Dec. 1[st] 2021, and is either 1.5x or 2x more transmissible than delta. These scenarios were further conditioned on the expectation that immunity will wane over 2.5 years.

The results of the forward simulations for these scenarios are shown in Fig 7A. These highlight that our model updated to data recorded to Sept. 24[th] 2021 is able to predict the rise of the large omicron wave observed for Florida from mid Dec. 2021. However, the predictions for the observed omicron wave that occurred was highly sensitive to the future first emergence date as well as the transmission rate of the modelled new variant. The best fitting model in this regard was the one that incorporated a Nov. 1[st] 2021 emergence date and a transmission rate that was 2x higher than that estimated for the delta variant (lowest RMSE value of 2338), followed by the model in which the variant was simulated to emerge on Dec. 1[st] 2021 at the same transmission rate (RMSE: 2913) (Table 4). These results indicate that while our sequentially fitted model that included a mechanism to counter parameter depletion has sufficient variance and capacity to predict the impact of future variants, it requires the timely provision of data on

**Table 4. RMSE and relative error of various omicron scenarios.**

| Scenario | RMSE |
| --- | --- |
| Nov. 1[st] 2021 Emergence, 1.5x Transmission Rate | 6002 |
| Nov. 1[st] 2021 Emergence, 2x Transmission Rate | **2338** |
| Dec. 1[st] 2021 Emergence, 1.5x Transmission Rate | 6362 |
| Dec. 1[st] 2021 Emergence, 2x Transmission Rate | 2913 |

the characteristics of these variants in order to generate reliable forecasts. Given that the best prediction was afforded by the November 1st emergence model, the data also suggests that a lead time of at least 6 weeks is required for the provision of such information to allow the reliable making of such a prediction.

Encouraged by the predictability of the Nov. 1st 2021/2x more transmissible model for the omicron wave, we inspected the impact of another future 5th more transmissible (3.5x and 5x more transmissible than delta) variant emerging in Florida on Mar. 1st, May 1st, July 1st and Sept. 1st 2022 on the course of the pandemic using this model. The results of these simulations are shown in Fig 7B, and provide two major insights. First, it shows that while outbreak size will depend on transmissibility of the new variant with higher transmissibility giving rise to bigger outbreaks, their overall sizes will be much smaller than the omicron wave. This is primarily due to the establishment of high levels of moderately long-lasting (over a duration of 2.5 years) population immunity in the population from both vaccinations and from previous waves, including the omicron wave. The second finding is that although varying complexly with changes in the fraction of the population that revert to being susceptible as a result of waning of immunity, there will be a tendency for these waves to become comparatively larger in size the longer out in the future a new variant arrives (best observed for the higher transmissibility modelled here; solid lines in Fig 7B). This is primarily due to the long-term slow growth in susceptibles as immunity wanes.

## Discussion

While there are understandable expectations among both the public and governments that vaccinations may finally portend the end of the COVID-19 pandemic, our data-driven modelling results reported here show that the pandemic could in fact follow different future paths depending on how rates of vaccinations may interact with variants, levels of social mitigation measures followed by a community, and critically on the effectiveness and durations of the immunity generated by the current vaccines in a population. If the population immunity to SARS-Cov-2 generated by vaccinations and from infections that had occurred in Florida operates over the long-term, then one future for the pandemic in the state with continuation of current levels of vaccination and social distancing measures is for cases to decay steadily until the pandemic fades out or ends with a high probability (Fig 5). Indeed, if a 99% fade out probability is used, we estimate that this would have occurred as early as around Nov. 13 2021. We also show that under such long-term immunity, a full release of social measures from Sept. 24th 2021, irrespective of whether vaccinations are maintained at the current rate or increased 1.5x (Fig 2), will no longer result in large increases in cases as would have been the case if such a release had occurred earlier in 2021 (Mar. 2021) because of the steady increase in this type of immunity over time (Fig 2). Increasing the vaccination rate to 1.5x (for example by vaccinating school-age children [45–47],while maintaining the estimates of social measures as of Sept. 24th 2021 into the future will also have only a little impact on this declining future path of the pandemic largely because a high level of population immunity (90%) had already been evolved by Sept. 24 2021 in the state (Fig 2). Essentially, these outcome patterns are also forecasted for hospitalizations and deaths (Fig 3 and Table 2), with only small increases in these outcomes predicted for a full release of the currently observed social measures irrespective of increased (1.5x) or whether the current vaccination rate is followed into the future.

Our forecasts for the course of the pandemic if immunity were to wane, however, project a dramatically different future path for SARS-CoV-2 transmission (Fig 4). Such a future will be complex but essentially the pandemic dynamics will be characterized by damped oscillations or formation of repeat waves of infection with the size of repeat infections and the inter-wave

periods or periodicity of the oscillations depending on how fast immunity wanes [48–50]. Faster waning could lead to sizeable infection waves and shorter inter-wave periods into the future, but over the long-run the pandemic will shrink in size and tend towards an endemic steady state (Fig 4). Our simulations further indicate that these effects will be accentuated if all social protective measures are fully released at daily vaccination rates as of Sept. 24th 2021, but that if the social measures alongside present vaccinations are continued then it is possible to not only curb peaks of the repeat waves but also lengthen the inter-wave period. If immunity were to wane over a relatively long period of time (5 years in our simulations), then the later interventions could even be optimal in curbing the oscillatory dynamics significantly to allow practical control of the pandemic (Fig 4).

At the time of writing, it is still not clear how long immunity to SARS-CoV-2 lasts although it is becoming apparent that immunity is likely to wane [18–20]. However, as shown in Table 3, the best fit to the Aug. 2021 peak cases observed in Florida in our simulations is provided by the model characterized by a moderately-long (2.5 years) duration of immunity, indirectly supporting the above findings from empirical studies that the overall population-level immunity (from both vaccinations and natural infections) generated to SARS-CoV-2 is likely to wane but at a rate that may not cause too rapid a decline in the achieved immunity. There is also growing evidence, in this connection, that the effectiveness (and duration) of immunity from vaccinations may differ from that induced by natural infections [51]. Such differences, if true, could indeed be driving the present post-vaccination resurgence in cases observed for US states that have achieved the highest vaccination rates relative to those that are yet to attain such levels, such as Vermont (https://www.nytimes.com/interactive/2021/us/vermont-covid-cases.html). We contemplate future work addressing these differences for the course of the pandemic, including assessing the optimal strategy (eg. introduction of 3rd booster vaccinations with or without minimal social mitigation measures [20]) for curbing any detected oscillatory dynamics in the transmission of the virus in different control settings. Note that these impacts of waning immunity could in reality also mean that policy makers might need to consider tuning and instituting repeat measures, including retaining some of the least socially disruptive social measures, to prevent the repeated flare-ups of the pandemic over a foreseeable future until some steady endemic state in viral transmission is reached. As noted, such permanency in responses may be seen as representing a new post-pandemic normal as it essentially involves fundamental longer-term changes to how a society functions normally such that viral transmission over the near-term future is contained within levels that may be safely tolerated [52].

It is also instructive to note that compared to permanent immunity, the probability of fade out of the pandemic is unlikely to reach high levels if immunity were to wane (Fig 6). Our simulations for a 2.5 year waning period indicates that this probability might increase in a stepwise fashion with each subsequent future wave, but will not reach 100% over the long-run. We also show how such calculations can be used along with information regarding longevity of immunity and future expected changes in compliance with social protective measures to indicate how the pandemic could play out in societies. Thus, step-wise changes in fade out probabilities that will not reach 100% over the long-term can be used to infer that the pandemic will not end but will settle into a wave-like behaviour in the future. Indeed, we could even make these calculations at a given time during the course of the pandemic to determine if a population is still at high risk for future resurgences. For example, our calculations of the fadeout probability for the pandemic made on Sept. 24 2021 indicated that it was close to 82.4% if immunity was permanent whereas it only 20.4% at that time if immunity waned over 2.5 years. This indicated that Florida was still very vulnerable following the delta wave and any dropping of social protective measures would put it at high risk to the omicron variant that emerged in the state from November 2021.

We have largely focused on the dynamics of infections in this study, although we show that if immunity is permanent, both hospitalizations and deaths will decline in the future under estimated vaccination and social mitigation rates as of Sept. 24[th] 2021 (Fig 3). A full release of social measures will result in an increase in both variables slightly in the immediate future after which both will again tend to fade alongside infections. While we could expect both variables to increase perhaps significantly above these levels if immunity were to wane rapidly as a result of the evolution of large repeat infection waves (Fig 4), recent data suggests that mortality rates may be declining in relation to infection levels overall owning to clinical as well as improvements in hospital care, increased testing, roll out of vaccinations, and possibly due to reduction in infective doses of the virus [53, 54]. This decoupling of deaths/hospitalizations from infection cases raises another possible future post-pandemic normal, viz. that societies could learn to live with a controlled level of transmission going forward via both the use of repeat vaccinations and the use of newly emerging therapeutics for managing disease outcomes [52]. In this scenario, long or fat-tailed risks [55] could be managed by targeting control (via temporary social distancing measures and/or targeting vaccinations to unvaccinated individuals) to emerging high-risk settings or sub-groups. Such post-pandemic normal strategies, however, will require implementing strong spatially explicit surveillance systems for tracking emerging cases as well as variants, and evolving adaptive management structures and capacities within health systems [22], which may be possible in settings with advanced, well-resourced, public care institutions but may prove challenging for less developed populations. Note that instituting such long-term digital surveillance can, however, also result in the erosion of personal freedoms and agency of individuals [56, 57], calling for careful analysis of the broader societal costs and benefits arising from the widespread deployment of these powerful technologies.

The evaluation of the capacity of our data-driven model based on sequential calibration to data for capturing information on initial conditions and temporal changes in external drivers of transmission while retaining parameter variance (see Methods), for forecasting the impact of novel variants has shown that if effectively leveraged such a modelling system can be used to simulate the outcomes of these type of new events reasonably well. However, as shown for the capacity of our model fitted to data to Sept. 24[th] 2021 for predicting the omicron wave (Fig 7A), this ability was highly sensitive to receiving timely information on date of first emergence as well as the transmission rate of a putative new variant. Indeed, given a plausible mix of emergence dates and virus transmissibility rates, inference can also be made regarding the likely impacts of variants arriving post-omicron (Fig 7B). We indicate in this regard that although new waves will be generated by more transmissible future variants their sizes will be governed by the duration and strength of prevailing immunity as well as the actual transmissibility rates of these variants. We also indicate that the longer out the new variant arrives the larger the size of the wave that will likely emerge (Fig 7B). These results demonstrate that our data-driven simulation model can be combined with evolving information regarding new variants to allow both quantitative forecasts and inferences on their likely impacts. One caveat, however, is that if a new variant is also more immune evasive, then it could increase the size of the future waves. The precise increase will need to be investigated in relation to the dynamics of naturally acquired vs. vaccine-induced immunity as well as their interactions with social measures to obtain a fuller description of such immune evasiveness on the future of the pandemic.

Overall, thus, our data-driven forecasts for the future course of SARS-Cov-2 in Florida indicate that contrary to the expectation that the introduction of vaccinations could lead to the permanent ending of the pandemic, additional futures could become possible if the immunity engendered through vaccinations and natural infections wane over time. Such futures will be

marked by repeated waves of infection, the amplitude and periodicity of which will depend on the duration over which the generated immunity in a population will operate. These complex futures will require recognition that continual vigilance and perhaps fundamental longer-term changes over the foreseeable future in both governmental responses and societal functioning as part of a new post-pandemic normal will be needed to control and mitigate against continuing outbreaks. A key current unknown that may confound these conclusions, however, is the period over which immunity to SARS-Cov-2 lasts [18–20]. Large repeat waves with short periodicity are possible with rapid waning of immunity, which will require strong control measures. We may be observing this already in US states, such as Vermont, that are observing large post-vaccination resurgence in cases despite high levels of vaccination. Another limitation of our work is that we use best-fitted models to project outcomes of various scenarios into the future. While the use of this combined predictive model/scenario approach can provide insights to possible future behaviors of the pandemic, it does not capture non-constant changes in future parameters related to interventions or virus transmission [58, 59]. Nor does it capture the reality that responses by policy-makers are often to the present incidence rate, which are likely to significantly influence the future course of the pandemic in complex ways [14]. Such scenario uncertainty may imply that it is not possible to formulate the probability of occurrence of one particular pattern *a priori*, suggesting that an ensemble of plausible but unverifiable scenarios might need to be simulated to understand how the pandemic may unfold with emerging data used to distinguish between such scenarios of future changes in key driving forces (such as followed in this work with regard to determining the likely duration of immunity (Table 3). Our data-driven projections are also dependent on the quality of data used to calibrate the present model through time. Anomalies in the data could bias model parameters and hence propagate errors in the projections, although the ensemble nature of our forecasting system takes account of these uncertainties to a large degree. In this work, we chose to neglect age stratification to restrict the number of parameters in the model. While the model is unable to predict age-specific outcomes, it can faithfully predict future cases and deaths at the aggregate level, especially in the short-term.

We conclude by noting that while we must always acknowledge the limitations surrounding the use of models calibrated to past data for precisely forecasting the futures of complex epidemics, such as COVID-19 [10, 11], our results do indicate that insightful model-based predictions of the future of the pandemic could be made if uncertainties about future changes in key drivers are captured appropriately through plausible scenarios, model parameter variance is maintained to a reasonable level over time, and if the plurality of the likely occurrence of possible futures is acknowledged and addressed in simulations. This suggests that provided these conditions are met, models, such as ours, can contain sufficient exchangeable information to predict the future path of the pandemic [60]. A key requirement for supporting such predictions, however, is the use of simulation analyses with timely observations of the outbreak and experimental data characterizing variants, both of which will be important for narrowing down which of the future predicted paths the present SARS-CoV-2 might follow.

## Supporting information

**S1 Fig. Average estimated transmission rate (black) and protection due to social measures (1-d parameter) over time.** The transmission rate is an averaged rate over alpha, delta, and all other variants. The priors on the d parameter are informed by Google Trends search data, as described in the main text.
(TIFF)

**S2 Fig. Daily reported vaccination rate in the state of Florida, as reported by coronavirus. app (https://coronavirus.app).**
(TIFF)

**S3 Fig. Proportion of alpha (red), delta (blue), and all other variants in the United States over time, as reported by the Helix COVID-19 surveillance dashboard (https://www.helix. com/pages/helix-covid-19-surveillance-dashboard).**
(TIFF)

**S4 Fig. Cumulative confirmed cases in Florida.** The median model prediction given estimates of social measures and vaccination rate as of Sept. 24th, 2021 is given in black, while 1 yr, 2.5yr, and 5yr immunity waning periods are shown in red, green, and blue, respectively. The solid lines represent estimated vaccination rate, while the dashed lines represent a 1.5x increase in vaccination rate.
(TIFF)

## Acknowledgments

This work was made possible by an internal grant from the University of South Florida. The funder had no role in study design, data collection and analysis, decision to publish, or preparation of the manuscript. A portion of the model runs was carried out using the MATLAB Parallel Computing Toolbox made available by USF Research Computing.

## Author Contributions

**Conceptualization:** Ken Newcomb, Shakir Bilal, Edwin Michael.

**Formal analysis:** Ken Newcomb, Shakir Bilal, Edwin Michael.

**Writing – original draft:** Ken Newcomb, Shakir Bilal, Edwin Michael.

**Writing – review & editing:** Ken Newcomb, Shakir Bilal, Edwin Michael.

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
