## [Decision Letter · Decision Letter 0]

25 Jul 2022

PONE-D-22-16941Combining predictive models with future change scenarios can produce credible forecasts of COVID-19 futuresPLOS ONE

Dear Dr. Michael,

Thank you for submitting your manuscript to PLOS ONE. After careful consideration, we feel that it has merit but does not fully meet PLOS ONE’s publication criteria as it currently stands. Therefore, we invite you to submit a revised version of the manuscript that addresses the points raised during the review process.

We look forward to receiving your revised manuscript.

Kind regards,

Martial L Ndeffo Mbah, Ph.D

Academic Editor

PLOS ONE

Journal Requirements:

Additional Editor Comments:

Thank you for submitting your paper to PLoS ONE. After peer review, it is our opinion that the manuscript needs major revisions before it can be deemed suitable for publication. Please, thoroughly address the reviewers comments as they would greatly improve the quality of the manuscript. Based on reviewer #3 comments, you may considering changing the manuscript title and some of the sections' title to better reflect your analyses and results.

Reviewers' comments:

Reviewer's Responses to Questions

**Comments to the Author**

1. Is the manuscript technically sound, and do the data support the conclusions?

Reviewer #1: Yes

Reviewer #2: Yes

Reviewer #3: No

2. Has the statistical analysis been performed appropriately and rigorously? 

Reviewer #1: Yes

Reviewer #2: Yes

Reviewer #3: I Don't Know

3. Have the authors made all data underlying the findings in their manuscript fully available?

Reviewer #1: Yes

Reviewer #2: Yes

Reviewer #3: Yes

4. Is the manuscript presented in an intelligible fashion and written in standard English?

Reviewer #1: Yes

Reviewer #2: Yes

Reviewer #3: Yes

5. Review Comments to the Author

Reviewer #1: The paper raises an important topic: can mechanistic SEIR fitted to data be used for long-term predictions of outbreak patterns/outcomes? The big confounding factor for Covid prediction is viral evolution shaped by complex interplay between within-host biology and environmental/behavioral factors, which brings novel stains with different epidemiological profiles.

Such evolutionary forecast is beyond current understanding and modeling tools. So the paper attempts a partial answer, by combining SEIR with putative future scenarios and timely inputs on ‘novel’ strain infectivity/transmissibility profiles. The model employs case–data from Florida, and highlights multiple SEIR compartments with focus on vaccination status and schedules, and the follow-up pathways.

Large part analysis is retrospective; it involves the outbreak data through Sep 2021 (delta-variant) to simulate predicted outcomes under different vaccine/prevention scenarios (Fig. 2-8). Its primary use is demonstrating the methodology. The more relevant part includes analysis of the latest omicron wave, and possible future developments (Fig 9).

While overall exposition is satisfactory, some parts need further elaboration and additional details.

Comments:

1. General question on long-term prediction simulations. The model fitting procedure (to my understanding) involves ‘dynamic parameters’ aggregated in 10-day blocks, and weekly averaged data, in particular variable transmission rates. The key question (needs clarification) is how such dynamic (variable) parameters are projected in the future.

2. A related question (needs elaboration): what are the primary sources of predicted variability (uncertainty envelops shown in figs 2-8), and how they relate to calibrated parameters, social/behavioral inputs, vaccine uncertainties.

3. Recovered (R) compartment and the role of immunity. Based on Fig.1 hosts could lose immunity (move to VC); and immune loss rate appears in the analysis. But model equations (GitHub DE) show no loss from R (needs clarification?)

4. Fig.1: multiple susceptible/vaccinated pools undergo similar SEIR- stages and transitions with asymptomatic/symptomatic pathways. Those pathways (and transitions) should depend on host’s immune status (needs more details). In addition, I’d like the authors to comment on possible effect of ‘vaccine’ and ‘natural immunity’ (R) on transmission/ disease pathways.

5. Do fig.2-3 assume full (post-infection) immunity? How different vaccinated/immune compartments (Fig. 1) combine in fig 2?

6. Elaborate uncertainty envelops of Fig.5.

7. Re fig 6 comments on ‘long-term oscillatory dynamics’ (repeated waves). Such patterns (convergence to endemic state via damped-oscillations) typically arise in SEIR models with immune loss.

8. Omicron wave (Fig. 9) employs enhanced transmissibility (factor 1.5-2 vs. delta). Does model account for reduced cross-immunity and possible breakthrough in different vaccinated classes subjected to omicron?

Minor comments

1. Multiple histories of fig 7-9 could be combined into ‘uncertainty envelops’, like in 2-6.

Reviewer #2: The manuscript "Combining predictive models with future change scenarios can produce credible forecasts of COVID-19 futures" by Newcomb analyses SARS-CoV-2 pandemic in the State of Florida. The authors develop an extended SEIR compartmental model that they calibrate to data, and then use to investigate future scenarios. Such predictions are important to anticipate the pandemic evolution, in order to timely change and adapt social measures. The authors study the impact of vaccinations, the consequences of immunity waning, and the impact of possible new variants. They conclude that new variants and loss of immunity may induce repeated waves of infection, but the severity of these waves is probably small due to a high levels of immunity acquired by vaccinations and natural infections.

In general, the study is interesting and well written. However, I have some comments that the authors might want to address in order improve the comprehension of their work, see pdf attachment

Reviewer #3: As the title anounces it, the authors claim that they are able to produce credible forecasts of COVID-19 futures. My opinion is that this is not the case, and the present paper does not add really useful material to the great mass of publications on this pandemic.

Of course, any reasonable model can be adjusted to past data, and varying several parameters allows to produce a variety of future predictions. That is essentially what is done in this paper, and it is far from providing realistic predictions. The statement that if immunity is permanent, there is a high probability of extinction of the pandemic in a new future, while in the case of waning immunity, it will continue for a long time, with a succession of waves, does not look as a very innovative prediction.

Two major problems about the future predictions concerning the pandemic are the loss of immunity, and the immune evasion by new variants. However, the second point is not seriously take into account, while the question whether or not some social distancing measures should be maintained, and whether or not the rate of vaccination should be multiplied by a factor 1.5 are presented as major options for controlling the future of the pandemic. These are not really serious options, since vaccination cannot be maintained at a high constant rate over a very long period of time, and most of the social distancing measures cannot be applied for ever.

Concerning the loss of immunity, the paper is very inconsistent from one section to another. In the middle of the second page of the final « Discussion » section, the authors recognize that « There is .. growing evidence .. that the effectiveness (and duration) of immunity from vaccination may differ from that induced by natural infection ». However, while it is written, 3 lines above table 1, that « the efficacy..of two-dose of mRNA vaccines.. decays to 67-80% after 7 months », the difference between « natural » and vaccine acquired immunity does not seem to be really taken into account. This 67-80 % is I think rather optimistic. What about after 10 or 12 months ? Moreover, many sections of the paper discuss the future of the epidemic in the hypothesis of permanent immunity, while others claim that a good estimate of the duration of immunity (without differentiating between natural and vaccine acquired immunities) is 2.5 years, an estimate which I think is extremely optimistic and irrealistic.

Table 1 presents vaccine efficacies used in the simulations. But those figures are absent from the discussion in the paper.

One subsection title at the end of the « Methods » section is far from what the paper does : « Fade out probability calculations » (what is done has nothing to do with any serious probability estimation/calculation). The next section « Estimation of Population Immunity.. » contains the following assertion : « This allowed us to estimate the change of levels of immunity due to natural infection versus arising from vaccination ». I do not see which serious procedure achieves this goal in the paper.

One serious drawback is the fact that, concerning new variants, the authors insist upon their date of emergence and their transmissibility, forgetting completely their immunity evasion, which seems to be a crucial point of Omicron for example.

One problem with this paper is that it seems to hesitate between drawing conclusions from the knowledge about the pandemic as it was towards the end of 2021, and inclusion of what has been learned since.

The paper contains twice the expression « exchangeable information », the second time at the end of the discussion section, where David Aldous’ St Flour Lecture Notes on Exchangeability is quoted. I do not understand what the authors mean by that expression, and how it might connect to Aldous’ lectures.

For all these reasons, I recommend rejection of this paper.

6. PLOS authors have the option to publish the peer review history of their article (what does this mean?). If published, this will include your full peer review and any attached files.

Reviewer #1: **Yes: **David Gurarie

Reviewer #2: No

Reviewer #3: No

---

## [Author Response · Author response to Decision Letter 0]

30 Sep 2022

We have attached a separate Microsoft Word file containing our point-by-point responses to each of the reviewer comments.

---

## [Decision Letter · Decision Letter 1]

31 Oct 2022

Combining predictive models with future change scenarios can produce credible forecasts of COVID-19 futures

PONE-D-22-16941R1

Dear Dr. Michael,

We’re pleased to inform you that your manuscript has been judged scientifically suitable for publication and will be formally accepted for publication once it meets all outstanding technical requirements.

Kind regards,

Martial L Ndeffo Mbah, Ph.D

Academic Editor

PLOS ONE

Additional Editor Comments (optional):

Reviewers' comments:

Reviewer's Responses to Questions

**Comments to the Author**

1. If the authors have adequately addressed your comments raised in a previous round of review and you feel that this manuscript is now acceptable for publication, you may indicate that here to bypass the “Comments to the Author” section, enter your conflict of interest statement in the “Confidential to Editor” section, and submit your "Accept" recommendation.

Reviewer #1: All comments have been addressed

Reviewer #2: All comments have been addressed

2. Is the manuscript technically sound, and do the data support the conclusions?

Reviewer #1: Yes

Reviewer #2: Yes

3. Has the statistical analysis been performed appropriately and rigorously? 

Reviewer #1: Yes

Reviewer #2: Yes

4. Have the authors made all data underlying the findings in their manuscript fully available?

Reviewer #1: Yes

Reviewer #2: Yes

5. Is the manuscript presented in an intelligible fashion and written in standard English?

Reviewer #1: Yes

Reviewer #2: Yes

6. Review Comments to the Author

Reviewer #1: The revised paper and the authors' responses addressed adequately my comments.

I defer to other reviewers to assess theirs

Reviewer #2: (No Response)

7. PLOS authors have the option to publish the peer review history of their article (what does this mean?). If published, this will include your full peer review and any attached files.

Reviewer #1: **Yes: **David Gurarie

Reviewer #2: No

---

## [Editor Report · Acceptance letter]

4 Nov 2022

PONE-D-22-16941R1 

Combining predictive models with future change scenarios can produce credible forecasts of COVID-19 futures 

Dear Dr. Michael:

I'm pleased to inform you that your manuscript has been deemed suitable for publication in PLOS ONE. Congratulations! Your manuscript is now with our production department. 

Kind regards, 

on behalf of

Dr. Martial L Ndeffo Mbah 

Academic Editor

PLOS ONE